# Single-Cell RNA-Sequencing Reveals Epithelial Cell Signature of Multiple Subtypes in Chemically Induced Acute Lung Injury

**DOI:** 10.3390/ijms24010277

**Published:** 2022-12-23

**Authors:** Chao Cao, Obulkasim Memete, Yiru Shao, Lin Zhang, Fuli Liu, Yu Dun, Daikun He, Jian Zhou, Jie Shen

**Affiliations:** 1Center of Emergency and Critical Medicine in Jinshan Hospital of Fudan University, Shanghai 201508, China; 2Research Center for Chemical Injury, Emergency and Critical Medicine of Fudan University, Shanghai 201508, China; 3Key Laboratory of Chemical Injury, Emergency and Critical Medicine of Shanghai Municipal Health Commission, Shanghai 201508, China; 4Fudan University Shanghai Medical College, Shanghai 200120, China; 5Shanghai Key Laboratory of Lung Inflammation and Injury, Shanghai 200032, China; 6Department of Pulmonary and Critical Care Medicine, Shanghai Respiratory Research Institute, Zhongshan Hospital, Fudan University, Shanghai 200032, China

**Keywords:** epithelial cells, developmental trajectories, single-cell RNA-sequencing, SOX9-positive AEC2s, chemically induced acute lung injury

## Abstract

Alveolar epithelial cells (AECs) play a role in chemically induced acute lung injury (CALI). However, the mechanisms that induce alveolar epithelial type 2 cells (AEC2s) to proliferate, exit the cell cycle, and transdifferentiate into alveolar epithelial type 1 cells (AEC1s) are unclear. Here, we investigated the epithelial cell types and states in a phosgene-induced CALI rat model. Single-cell RNA-sequencing of bronchoalveolar lavage fluid (BALF) samples from phosgene-induced CALI rat models (Gas) and normal controls (NC) was performed. From the NC and Gas BALF samples, 37,245 and 29,853 high-quality cells were extracted, respectively. All cell types and states were identified and divided into 23 clusters; three cell types were identified: macrophages, epithelial cells, and macrophage proliferating cells. From NC and Gas samples, 1315 and 1756 epithelial cells were extracted, respectively, and divided into 11 clusters. The number of AEC1s decreased considerably following phosgene inhalation. A unique SOX9-positive AEC2 cell type that expanded considerably in the CALI state was identified. This progenitor cell type may develop into alveolar cells, indicating its stem cell differentiation potential. We present a single-cell genome-scale transcription map that can help uncover disease-associated cytologic signatures for understanding biological changes and regeneration of lung tissues during CALI.

## 1. Introduction

Chemically induced acute lung injury (CALI) is caused by chemical gas poisoning, smoke inhalation injury, or reflux of gastric juice, with chemical gas poisoning being the most common cause [1]. CALI leads to the chemical production or transport of toxic gaseous chemicals, such as phosgene, diphosgene, chlorine, ammonia, and hydrofluoric acid [2]. Phosgene is used globally in industrial processes, including the synthesis of pesticides, plastics, dyestuffs, polyurethanes, and agrochemicals, and is indispensable in pharmaceutical production [3]. Thus, accidental, environmental, and occupational exposure to phosgene cannot be avoided, resulting in a high incidence of CALI [4]. Cases of accidental or intentional exposure to the aforementioned chemicals that require treatment have been increasing annually as large chemical stockpiles still exist; however, effective therapies are currently lacking [5].

Inhalation exposure to phosgene is a common cause of CALI, resulting in direct damage to the lung tissue and leading to injury, the severe effects of which are characterized as acute respiratory distress syndrome (ARDS). Although supportive therapies are available, no effective methods have been developed to reverse the physiological damage triggered by phosgene [6,7]. In most acute cases of phosgene inhalation exposure, both upper and lower airways are affected, causing injury, impaired gas exchange, and epithelial sloughing [8], ultimately leading to ARDS.

Alveolar epithelial cells (AECs) have been implicated in the pathophysiology of CALI [9,10], suggesting that impaired epithelial regenerative capacity is a key event contributing to the pathogenesis of CALI. Multiple progenitor cell populations differentiate into alveolar epithelial type 2 cells (AEC2s) [10,11,12,13] during epithelial injury. After an injury, alveolar epithelial type 1 cells (AEC1s) are regenerated via differentiation from AEC2s. The alveolar epithelium is maintained and repaired via AEC2 proliferation and hyperplasia and then differentiated into AEC1s. Researchers have attempted to restore impaired tissue function after lung injury. However, this requires a thorough understanding of the distribution of physiological tasks among cell types and the change occurring in cell states during homeostasis, injury/regeneration, or disease. Cell type-specific molecular signatures or specific subsets of stem cells can help in implementing a regenerative therapeutic approach for pulmonary diseases [14]. However, the cell homeostasis or molecular mechanisms that induce AEC2s to proliferate, exit the cell cycle, and transdifferentiate into AEC1s are poorly understood.

This study aimed to investigate the epithelial cell types and states in phosgene-induced CALI rat models using single-cell RNA-sequencing (scRNA-seq) of bronchoalveolar lavage fluid (BALF) samples and sequencing data of epithelial cells to explore progenitor cells that may regenerate into AECs. Our results will substantially influence CALI therapy with epithelial cell repair and promote lung regeneration after injury.

## 2. Results

### 2.1. Cell Sequencing and Clustering to Identify Main Cell Types

We performed scRNA-seq of cells isolated from all subpopulations of the BALF (Figure 1a). The results showed that all cells were evenly distributed in each cell cluster, and the individual samples were merged (Appendix A). Through unbiased graph-based clustering, 23 major cell populations were identified (Figure 1b). Each cell was evenly distributed in each cell cluster; the t-distributed stochastic neighbor embedding (tSNE) plots of the expression and distribution of marker genes in epithelial cells, macrophages, and macrophage proliferating cells are presented in Appendix A. The differences in the abundance of these cell states across the phenotypes from different samples were further analyzed (Appendix A), and the expression of canonical markers by each cell type was visualized (Appendix A).

As we aimed to identify the mechanisms of physiological repair by AECs, a specific marker of epithelial cells, EpCAM, 4 clusters (8, 18, 20, and 22 those expressing EpCAM) were selected for further study (Figure 1b). We profiled total epithelial cells (800–1500 cells per sample) using a 3′ RNA tag sequencing approach (Appendix A). Next, we extracted lung epithelial cells for further analysis. We found 3071 epithelial cells by identifying *Epcam*. The epithelial cells could be divided into 11 different clusters after dimensionality reduction (Figure 1b and Appendix A). The top four marker genes of epithelial cells in clusters 8, 18, 20, and 22 were highly expressed in epithelial cells (Appendix A), indicating the reliability of the results. We identified 4744 differentially expressed genes (DEGs) and 18 signature genes that distinguished in each subpopulation (Figure 1c), and then visualized and classified the annotation of cell types (Figure 1d). A heatmap was generated using the marker gene expression in these 11 cell clusters (Figure 1e), based on the expression of the top 10 most significant DEGs in each cell cluster. Alveolar SFTPC, a unique protein of AEC2s, can manifest the stem cell attribute subgroup in epithelial cells. Based on the detected gene signatures, we concluded that the targeted population mainly comprised AEC2s expressing typical proteins such as EpCAM, SFTPC, and SOX9 (Figure 1f,g).

### 2.2. SOX9-Positive AEC2 Differentiation and Maturation Trajectories

We also conducted hierarchical clustering (Figure 1h). Using Monocle [15], which places differentiating epithelial cell subsets along granulopoiesis trajectories in pseudo-time (Figure 1k), AEC differentiation and maturation were found to occur on a tightly organized trajectory, starting from clusters 0, 3, and 8 (SOX9-positive cells) in BALF and ending with clusters 6 and 7 in cells in the CALI state (Figure 1i and Appendix A). Combined with the results of the pseudo-time analysis and the targeted annotation of epithelial cell subsets, these results suggest that SOX9-positive AEC2s, similar to stem cells, can differentiate in multiple directions. The top 44 genes were found to affect the fate of cells based on the detailed genetic information obtained (Figure 1j). Furthermore, *Erich4*, *Rn60-20-0066.3*, *Rprml*, and *Sh2d7* were the top four genes that affected cell fate (Figure 1l,m). These genes may play an essential role in the differentiation of SOX9-positive AEC2 cell.

### 2.3. Annotation of Pulmonary Epithelial Subtype Based on the Marker Gene

We then performed downstream analysis after data integration and annotated epithelial clusters with the marker genes that were previously reported and validated (Table 1) using tSNE and UMAP plot analyses (Figure 2a). We identified major epithelial cell distributions, including AT_Sox9, AT_Upk3bl, Cillated_Cfap45, Club_Scgb3a2, and Fibroblast_Igf1 cells, based on the expression of marker genes in each cluster of epithelial cells (Figure 2b). Further, we compared the cell population of each cluster and visualized the proportion of cells in each cell cluster in the normal control (NC) and phosgene-induced CALI model (Gas) groups (Appendix A). As shown in the Roseplot map in Figure 2c, the percentage of SOX9-positive AEC2s increased to 6.15% in Gas compared with 3.8% in NC. Furthermore, the number was significantly increased compared with that under normal conditions (Figure 2d). The cluster of SOX9-positive AEC2s containing signature genes was profoundly different from the clusters of other cell types (Figure 2e), indicating a special state of AECs. Further analysis confirmed the distinct gene expression patterns among the five clusters (Figure 2f and Appendix A). We analyzed the typical expression of genes in each cluster (Figure 2g) and identified the first four marker genes, that is, *Sh2d5*, *Cldn18*, *Pik3cg*, and *Lrmp* (Figure 2h) in SOX9-positive AT cells and determined the highest percentage of the four marker genes *Rprml, Rxrg*, *Crym*, and *Tmem229a* (Figure 2i). It has been suggested that SOX9 plays multiple roles in the lung epithelium during branching morphogenesis [16], and SOX9-expressing cells may serve as therapeutic targets in lung tissue after radiation-induced lung injury [17]. However, an in-depth understanding of the contribution of these cells, especially SOX9-positive cells, to CALI, is still lacking.

### 2.4. Proliferation of SOX9-Positive AEC2s in the Bronchoalveolar Area in CALI Rats

To confirm the function of SOX9-positive AEC2s identified using scRNA-seq, we measured the changes in the quantity of these cells and functional expression in the CALI state. The lung tissues of the NC group exhibited normal structures without histopathological changes, and extensive and severe pulmonary edema and high ALI scores were observed in the lung tissues of the Gas group (Appendix A). Furthermore, the *Sftpc* and *Sox9* transcript levels significantly increased in the total lung transcripts in the Gas group compared with those in the NC group. In contrast, the *Epcam* transcript levels showed a significant decrease in the total lung transcripts (Figure 3a and Appendix A), suggesting the impairment of AECs. Analysis of lung sections coimmunostained with antibodies against SOX9 and SFTPC revealed a notable increase in the number of SOX9 and SFTPC dual-positive cells in the Gas group lungs compared with those in the NC group lungs. However, there was scarce or no staining in the SOX9 or SFTPC-positive AEC2s in the developing lungs of the NC group (Figure 3b). The identity of the sorted SFTPC and SOX9 cells was verified by measuring the expression of SOX9-positive AEC2 marker genes. Using this method, we determined the percentage of the targeted subpopulation in the lung tissues. This result is consistent with the scRNA-seq results. Analysis of the coimmunostained lung sections showed a marked increase in the number of SOX9 and SFTPC dual-positive cells in the Gas group lungs compared with those in the NC group lungs (Figure 3c). The expression of the proliferation markers Ki67 and PCNA of SOX9-positive cells was significantly upregulated in the Gas group compared with those in the NC group (Figure 3d,e); this finding corroborated the scRNA-seq result. Immunohistochemical analysis revealed that SOX9 is predominantly localized at the nuclei of spindle-shaped cells around the bronchoalveolar area, where AEC2s are predominantly distributed. Quantification of the number of SOX9- and SFTPC-positive cells in these lesions indicated a marked increase in the percentage of SOX9-positive AEC2s in the Gas group compared with that in the NC group (Figure 3f,g).

To show that SOX9 expression was upregulated in AEC2s, we determined the levels of relevant proteins using lysates of lung AECs isolated from the NC and Gas groups. We observed a significant increase in the levels of SFTPC and SOX9 as well as the level of thyroid transcription factor-1 (TTF-1), another specific marker of AECs, in the total protein of tissues isolated from CALI lungs compared with those isolated from normal lungs. Similarly, the EpCAM level was significantly decreased in CALI lungs, indicating damage to the epithelial cells (Appendix A). Overall, in the lung injury model induced by phosgene inhalation, SOX9-positive AEC2s proliferated and were mainly present in the alveolar area. In addition, the gene and protein levels of Sftpc and Sox9 were significantly upregulated compared to normal homeostasis, consistent with the scRNA-seq results.

### 2.5. scRNA-seq Uncovers DEGs Predominantly Expressed in AECs during CALI

We analyzed the DEGs in SOX9-positive AEC2s between the groups to explore the major transcriptomic changes in the NC and Gas groups. The expression of *Rgs5* and *Spid* was significantly upregulated throughout the clusters of other cells, whereas they were expressed only in SOX9-positive cells in the normal state (Figure 4a and Appendix A). Based on the expression of the most significant DEG in the AEC cluster, we generated a heatmap of marker gene expression (Figure 4b) and compared the major transcriptomic changes in SOX9-positive AEC2s. Fifteen genes were significantly upregulated, whereas 10 were significantly downregulated (Figure 4c). Gene Ontology (GO) analysis was performed using all DEGs to understand further the transcriptomic differences between the NC and Gas groups. The DEGs were enriched in the AEC-mediated actin cytoskeleton, posttranscriptional regulation of gene expression, and actin-binding (Figure 4d–f). The Kyoto Encyclopedia of Genes and Genomes (KEGG) pathway analysis showed that DEGs were mostly enriched in the PI3K-Akt signaling pathway (Figure 4g), indicating a possible functional pathway. Finally, the identified signaling pathways were found to be involved in regulating the pluripotency of stem cells (Figure 4h and Appendix A) and cell cycle (Figure 4i), as well as the citrate cycle (Appendix A), base excision repair (Appendix A), and mismatch repair (Appendix A). These results provide the direction of cell differentiation and function, which warrant further studies.

## 3. Discussion

In this study, we performed single-cell transcriptome profiling to reveal AEC heterogeneity, orchestrated maturation during homeostasis, and identified a specific state of AECs in the BALF, namely, SOX9-positive AEC2s, which are expanded during CALI. These cells are potential stem cells involved in repair and regeneration following CALI. This study helps in discovering new cell subtypes and explaining the development track of SOX9-positive AEC2 cells in adult lung tissue to identify stem cells, providing novel insights for improving treatment efficacy.

The incidence of phosgene-induced CALI has been increasing in recent years. CALI directly damages the respiratory tract and alveolar tissue. Therefore, it is necessary to restore normal organ homeostasis by regenerating the functional tissue after severe trauma [18]. Multiple stem/progenitor cell populations play essential roles in repairing the damage induced by different types of acute lung injuries [19]. However, the cell components between the airway and lung parenchyma as well as those within different parts of the airway are considerably different. For example, basal cells are distributed in the trachea and bronchi, and secretory cells are distributed in the bronchi and bronchioles [20]. Alveolar stem cells are found in the junction of the bronchioloalveolar tubes [21], whereas AEC2s are found in the alveolar area [22]. Type 1 (AEC1s) and type 2 (AEC2s) cells constitute the alveolar epithelium, and AEC1s, the most common cells (covering over 95% of the alveolar surface), provide efficient gas exchange because of their thin morphology. However, AECs, particularly AEC1s, undergo excessive apoptosis and necrosis, contributing to the pathogenesis of several acute and chronic lung diseases; we also confirmed considerable damage in AEC1s. Subsequently, AEC2s secrete surfactants to protect alveoli from collapsing and act as stem cells to regenerate AEC1s during damage repair [22,23]. In several severe diseases, including respiratory distress syndrome and idiopathic pulmonary fibrosis, the body cannot establish or maintain AEC2s and AEC1s [24]. AEC2 is an alternate progenitor cell type responsible for regenerating the injured alveolar epithelium; marked cell proliferation occurs while tissue damage is repaired.

To identify rare subpopulations of proliferating AEC2s that exit the cell cycle and transdifferentiate into AEC1s, we used scRNA-seq, which is ideally used for identifying subtypes within a heterogeneous population and providing perspective into the regulation of transitions between cellular states. We found that AECs are composed of five cell types and a unique subpopulation of epithelial progenitor cells characterized by high SOX9 expression coupled with high expression of specific markers of AEC2s. SOX9-positive distal-tip cells are lung progenitor cells that differentiate into airway and alveolar cells in the early pseudo-glandular stage of lung development, as shown using genetic genealogy tracing [17,25]. Nichane et al. grafted SOX9-positive distal-tip embryonic lung progenitors into a damaged adult mouse lung as a single-cell population. The authors found that the cells differentiated regionally into all major lung epithelial cell types, constituting both conducting airways and alveoli [26]. SOX9 and SFTPC are expressed in human lung bud tip progenitor cells, giving rise to alveolar cells in the late developmental stages [27]. A few cells (1 or 2 cells in each trachea) at the junction of the bronchioloalveolar duct were identified as SCGB1A1 (secretory cells); additionally, we identified alveolar SFTPC, a unique protein expressed by AEC2s [28]. Our study identified cell cluster 6 of SOX9-positive cells, whose SFTPC expression was high.

This is the first report of the expansion of SOX9-positive AEC2s within a CALI lung. Lung injury results in the activation of specialized epithelial progenitor cells to regenerate the epithelium. For example, activation of the innate and adaptive immune systems is hypothesized to lead to endothelial and epithelial cell injury, resulting in vasculopathy or the transformation into and expansion of myofibroblasts. Based on the severity of the injury, the remaining AEC2s and distal airway stem/progenitor cells mobilize to cover the stripped alveoli and restore the normal barrier. SOX9-positive AEC2s are relatively fewer or almost absent in the cluster under a steady state, but their numbers and percentages increase substantially following phosgene inhalation. Additionally, SOX9-positive AEC2 marker genes were considerably upregulated in signaling pathways regulating the pluripotency of stem cells, cell cycle, and base excision repair, indicating the high functionality of this subpopulation. Histopathological analysis suggested that these cells localize to areas of the bronchoalveolar junction and may be activated by the PI3K-Akt pathway. These cells may have a unique association with the epithelial–mesenchymal transition, where evolving cells have epithelial characteristics and develop AEC1 polarity during the transition from inflammation. By characterizing the epigenomic and functional phenotypes of the SOX9-positive AEC2 state, our study provides insights into its biology and supports a model where SOX9-expressing stem cells can be used for targeted tissue regeneration during CALI.

However, the study has some limitations. We analyzed AECs in CALI 24 h after inducing lung injury; however, it is important to explore other time points for dynamic monitoring, including the state of cells after differentiation. In addition, the specific mechanisms of SOX9-positive AEC2 cell repair or signaling pathways still deserve further exploration. We intend to conduct research from this perspective in the future. In summary, our study describes a cell state where SOX9-expressing AEC2s exhibit stem cell properties in pulmonary epithelial cells characterized by their epigenomic and functional phenotypes, imparting regenerative and repair capacities to AECs. This study demonstrates the utility of single-cell genomics in discovering disease-associated cytologic features and provides insights into the cellular basis of tissue repair during CALI.

## 4. Materials and Methods

### 4.1. Animals and Grouping

Six male Sprague–Dawley rats (weighing 200 ± 20 g, 6–8 weeks old; Experimental Animal Center, Naval Medical University, Shanghai, China) were raised in a climate-controlled cage (temperature, 24–26 °C; humidity, 55–60%; and dark/light cycle, 12/12 h). The rats were randomly and evenly divided into the normal control (NC, n = 3) and Gas (n = 3) groups. The rats in the Gas group were subjected to phosgene inhalation as previously described [29]. Briefly, the rats in the NC group were exposed to normal room air, whereas the Gas group was exposed to phosgene in an airtight cabinet at a final concentration of 8.33 mg/L phosgene for 5 min. Phosgene gas was produced by dripping N, N-dimethyl formamide (Macklin, Shanghai, China) into hexamethylene-containing triphosgene (Macklin). Animal protocols were approved by the Institutional Animal Care and Use Committee of Jinshan Hospital, Fudan University, China, and complied with the revised Animals (Scientific Procedures) Act 1986 in the UK and Directive 2010/63/EU in Europe.

### 4.2. Isolation and Preparation of Single Cells and Collection of Lung Tissues

After 24 h, BALF samples from rats were freshly procured for scRNA-seq. All samples were processed within 1 h after collection under the biosafety S3 conditions. The left lung tissues of rats in both groups were collected, frozen immediately, and fixed in 4% paraformaldehyde for over 24 h at 4 °C.

### 4.3. scRNA-seq

BALF cells were loaded into phosphate-buffered saline containing 0.05% bovine serum albumin following the 10× Genomics protocol [30]. The cell preparation time was <2 h; subsequently the cells were loaded onto the 10× Chromium controller. Libraries were constructed using the Single Cell 3Library Kit V2 (10× Genomics, Pleasanton, CA, USA). Transcriptome profiles of individual cells were determined using 10× Genomics-based droplet sequencing. The processed libraries were sequenced on an Illumina HiSEQ4000 platform using 26 × 8 × 101 sequencing. The scRNA-seq experiment was performed three times using a different NC and Gas group rat each time. The sequencing metrics, including depth and saturation, are shown in Appendix A.

### 4.4. Data Quality Control

The data were analyzed using the Seurat package of R for further data quality control (QC). Based on the median number of genes and percentage of mitochondrial genes in lung samples, cells were screened, and those with <200 and >3000 genes (potential cell diploids) and > 20% of mitochondrial genes were excluded [31]. After QC, 37,245 and 29,853 high-quality BALF cells in the NC and Gas groups, respectively, were obtained. The relationships between the percentage of mitochondrial genes and mRNA reads and between the quantity of mRNA and mRNA reads were examined and visualized. After normalizing the data, all highly variable genes in the included data were identified after controlling for the relationship between mean expression and dispersion. All variable genes were used for the downstream principal component analysis. The results showed that the first 50% had good discrimination.

### 4.5. Integrated Analysis

We performed canonical correlation analysis (CCA) [32] to examine the batch effects and integrate the data. The CCA basis vectors were aligned between the data sets to form a single, integrated low-dimensional space. For this analysis, the following parameters were adopted: top 2000 genes with the highest dispersion from 6 datasets (by default) and the first 50 canonical correlation vectors.

### 4.6. Unsupervised Clustering and Annotation

We used the FindClusters function (resolution: 0.3, 0.6, and 0.2) to control for the total cells. All cells, including epithelial cells, were primarily annotated according to their respective expression levels using the Louvain algorithm, combined with further clustering analysis of each cell subpopulation by combining classical cellular markers based on the same principal components as the RunUMAP function (Table 1). The major cells of the BALF, including macrophages, epithelial cells, and macrophage proliferating cells, were identified.

### 4.7. GO Term and KEGG Pathway Enrichment Analysis

GO enrichment analysis of DEGs corrected for gene length bias was performed using the cluster Profiler R software package (3.18.0). GO terms with corrected *p*-values < 0.05 were considered significantly enriched by DEGs. The statistical enrichment of DEGs in the KEGG pathways was examined using the cluster Profiler R package [33].

### 4.8. Identification of DEGs

The FindMarkers or FindAllMarkers function [test.use = ‘‘t”, logfc.threshold = log(1.5)] was used to identify DEGs according to normalized data. Based on the total number of genes in the dataset, *p*-value adjustment was performed using Bonferroni correction. DEGs with adjusted *p*-values > 0.05 were filtered out. GO analysis was performed using the clusterProfiler60 R package. The non-parametric Wilcoxon rank-sum test was used to conduct gene expression analysis and identify DEGs with an average fold-change >2.

### 4.9. Developmental Trajectory Inference

Pseudo-time trace was generated using Monocle version 2 to infer the potential lineage differentiation trajectory as previously detailed [15]. The newCellDataSet function (lowerDetectionLimit = 0.5; expressionFamily = negbinomial.size) was used to get the object based on the highly variable genes identified using Seurat version 2.3.4.

### 4.10. Reverse Transcription-Quantitative Polymerase Chain Reaction

The total RNA was extracted using the TRIzol reagent (RR047A; Takara, Japan). The cDNA was synthesized and used as the template for reverse transcription-quantitative polymerase chain reaction (RT-qPCR) performed using the SYBR Green Premix Pro Taq HS qPCR Kit (cat. no. AG11701; Accurate Biology). The qPCRs for *Sox9,* surfactant protein c gene (*Sftpc*), and epithelial cell adhesion molecule gene (*Epcam*) in technical triplicates and using three 10-fold dilutions of standardized genome equivalents were performed using the SuperScript III One-Step RT-PCR Kit (12574026; Thermo Fisher Scientific, Waltham, MA, USA) and the Roche LightCycler 480 platform. RT-qPCR was performed using an Eppendorf Realplex 4 instrument (Eppendorf, Hamburg, Germany). The sequences of the PCR primers were obtained from Univ-bio (Shanghai, China) and are provided in Appendix A. The relative expression of each gene was normalized to that of *Actb.*

### 4.11. Immunohistochemical Staining and Immunohistochemical Score

Histological procedures were performed for immunological analyses; 5-μm microtome sections were deparaffinized and subjected to Masson’s trichrome staining. Endogenous peroxidase activity in the sections was quenched with 3% H_2_O_2_, and the epitopes were thermally retrieved. The sections were incubated with primary antibodies specific for SFTPC and SOX9 at 4 °C overnight (Appendix A). The specimens were washed thrice with phosphate-buffered saline and incubated with secondary antibodies at 37 °C for 30 min. Enzymatic assays were performed using horse radish peroxidase-conjugated antibodies. The immunohistochemical score was calculated by multiplying the percentage of positive cells by the intensity of staining. The intensity was scored as 0 (negative), 1+ (weak staining), 2+ (moderate staining), or 3+ (strong staining), whereas the frequency was scored based on the proportion of positive cells.

For histological analysis, the lung tissue was fixed overnight at 4 °C in 4% paraformaldehyde, dehydrated, embedded in paraffin, and then cut to a thickness of 8 μm using a microtome (Leica, Wetzlar, Germany). Hematoxylin and eosin staining and Masson’s trichrome staining were performed according to standard protocols, followed by imaging under a light microscope (Nikon, Tokyo, Japan).

### 4.12. Immunofluorescence

The lung sections were immunostained, as described previously [17]. Briefly, the tissues were fixed in 4% paraformaldehyde and treated with Triton X-100 permeabilization buffer (Sigma-Aldrich, St. Louis, MI, USA). The samples were processed, embedded in paraffin, and cut into 5-μm thick sections. The lung sections were deparaffinized, hydrated, boiled in Target Retrieval Solution (Agilent, Palo Alto, CA, USA), blocked in 5% goat or donkey serum in Tris-buffered saline with 0.05% Tween, and incubated with antibodies against SFTPC, SOX9, Ki67, and PCNA (Appendix A) overnight at 4 °C. The samples were stained with DyLight 488 and Alexa Fluor 594 (Invitrogen, Waltham, MA, USA) for 1 h at room temperature. Subsequently, the cell nuclei were stained with DAPI at 1:1000 for 1 min. Images were obtained using a laser scanning confocal microscope (Carl Zeiss AG, Oberkochen, Germany). Data were analyzed using Imaris software (Imaris 8.1; Oxford Instruments, Abingdon, United Kingdom).

### 4.13. Western Blot Analysis

The collected lung tissue samples were lysed using a protein extraction kit (Well-bio, Shanghai, China) according to the manufacturer’s instructions. After transferring the samples into a blocking buffer for 2 h, membranes with different bands were probed with primary antibodies and incubated overnight at 4 °C. The lysates were electrophoresed and transferred onto nitrocellulose membranes. Antibodies and their dilutions used for immunostaining are listed in Appendix A. The densities of the bands on the membranes were scanned and analyzed using the phosphoimager software Multi Gage (Fujifilm, Tokyo, Japan). The expression of the target proteins was normalized to that of GAPDH.

### 4.14. Statistical Analysis

The scRNA-seq, with 3 NC and 3 Gas rats, was performed once. Cell type numbers in the multivariate case were compared using logit models, Tukey’s test, and Fisher’s exact test. Differential gene expression was reported using adjusted *p*-values with Bonferroni correction for multiple hypothesis testing. All data were statistically analyzed using GraphPad Prism 8.0 and are expressed as mean ± standard deviation. Significant differences between groups were analyzed using two-tailed unpaired Student’s *t*-tests. Statistical significance was set at *p* < 0.05.

## 5. Conclusions

In summary, our study describes a cell state in which SOX9-expressing AEC2s exhibit stem cell properties characterized by epigenomic and functional phenotypes in pulmonary epithelial cells, imparting regenerative and repair capacities to AECs. This study demonstrates the utility of single-cell genomics in discovering disease-associated cytologic signatures and provides insights into the cellular basis of tissue repair during CALI.

## Figures and Tables

**Figure 1 ijms-24-00277-f001:**
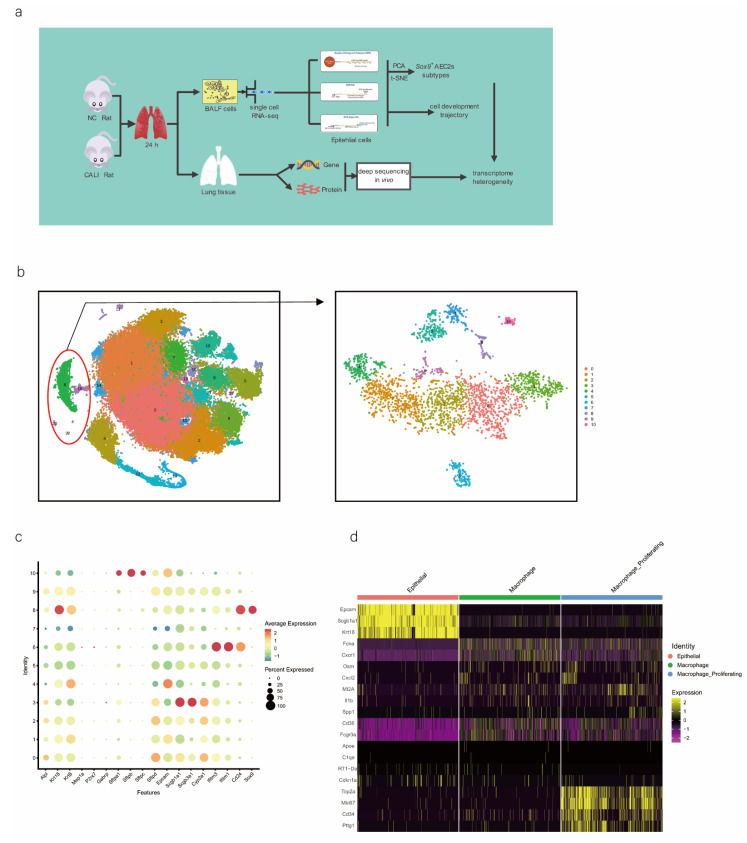
scRNA-seq of alveolar lavage cells isolated from rats in the normal control (NC) and phosgene-induced CALI model (Gas) groups. Sprague–Dawley rats were treated with phosgene to establish the Gas group. The NC group was not subjected to phosgene inhalation. After 24 h, alveolar lavage cells from both groups were loaded into a 10× Genomics Chromium Single Cell 3′ Solution. (**a**) Design of the study and processing pipeline for BALF samples and lung tissues used in this study. (**b**) tSNE plots of the 23 clusters of BALF cells. The scatter plot of classic marker genes (*Epcam*) presents the distribution of epithelial cells in the NC and Gas groups. (**c**) Dot plot of the scaled expression of selected feature genes in each cluster. The colors indicate the average expression of genes in each cluster scaled across all clusters. (**d**) Annotation of cell types was visualized and classified. (**e**) Heatmap of differentially expressed genes in each cluster. (**f**) Violin plots of the expression of marker genes in each cluster of epithelial cells; specific molecules were introduced by *Epcam*, *Sftpc*, and *Sox9*. (**g**) tSNE plots of the targeted subpopulation of cells, that is, SOX9-positive AEC2s. (**h**) tSNE plots showing subtype distribution and reconstruction of the developmental trajectory of epithelial cells. (**i**) Monocle trajectories of AECs colored by sample source (left) and cluster identity (right). Each dot represents a single cell. Cell orders are inferred from the expression of the most variable genes among all cells. The trajectory orientation was determined based on prior biological information. (**j**) Heatmap of the top 44 genes affecting cell fate decisions. The 44 genes are divided into six clusters. (**k**) Cells ordered by Monocle pseudo-time in each cluster of AECs. (**l**) The top four genes that could affect the fate of cells, indicating the change in pseudo-time. (**m**) The pseudo-time trajectory of epithelial cells in three different states.

**Figure 2 ijms-24-00277-f002:**
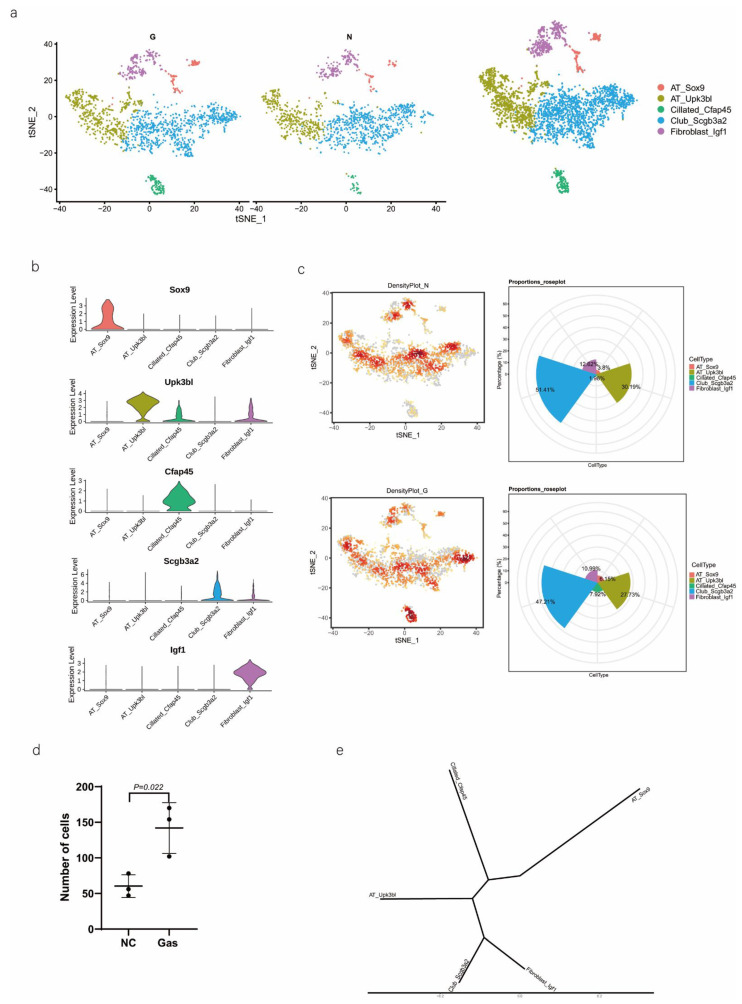
Identification of several types of epithelial cells by referring to the literature and the webtool CellMarker. (**a**) tSNE plots of the cell-type distribution of all identified epithelial cells, including AT_Sox9, AT_Upk3bl, Cillated_Cfap45, Club_Scgb3a2, and Fibroblast_Igf1 cells. (**b**) Violin plots of the expression of the signature genes in each cluster of epithelial cells. (**c**) Roseplot map of the proportion of each cell cluster in the NC and Gas groups. (**d**) Number of AT_Sox9 cells (SOX9-positive AEC2s). (**e**) Similarities in the signature gene expression in AEC2 clusters. Distance = (1 − Pearson correlation coefficient)/2. (**f**) Heatmap of the top 10 expressed genes of each cluster. (**g**) tSNE plots of the typical expression gene in each cluster. (**h**) tSNE plots of the top four marker genes in SOX9-positive AT cells. (**i**) Percentage of the top four marker genes.

**Figure 3 ijms-24-00277-f003:**
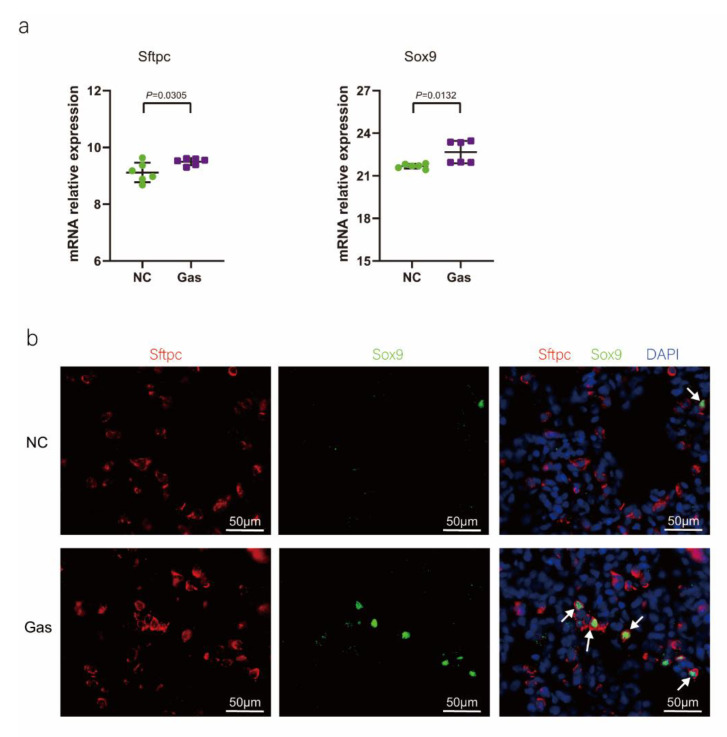
SOX9 expression is upregulated in lung AECs. (**a**) qPCR analysis of *Sftpc* and *Sox9* transcripts at different stages of lung development. (**b**) Lung sections of NC and Gas group rats stained for SFTPC and SOX9. Representative confocal images of lung sections stained for SFTPC (green color), SOX9 (red color), and DAPI (blue color). Cells positive for both SOX9 and SFTPC are highlighted with white arrows. Scale bar: 50 μm. (**c**) SOX9- and SFTPC-positive cells in the total cells quantified using the Elements image analysis software. (*p* < 0.001, n = 6, unpaired *t*-test). (**d**) Coimmunostaining of SOX9 and Ki67 and (**e**) coimmunostaining of SOX9 and PCNA in the distal lung AECs from NC and Gas group lung cultures. Representative images of SOX9-stained lung sections of NC and Gas group rats captured at 4× (low, original) magnification. Cells positive for both SOX9 and SFTPC or PCNA are highlighted with white arrows. Scale bars: 50 μm. (**f**) Representative immunostaining images of SFTPC in the lung sections of NC and Gas group rats obtained at 20× (left) and 40× magnification (right). Scale bar: 50 µm. Quantification of SFTPC-positive cells in the total cells. (*p* < 0.001, n = 4, unpaired *t*-test). (**g**) Representative immunostaining images of SOX9-stained sections obtained at 20× (left) and 40× magnification (right). Scale bar: 50 µm. Quantification of SOX9-positive cells in the total cells. (*p* < 0.001, n = 4, unpaired *t*-test).

**Figure 4 ijms-24-00277-f004:**
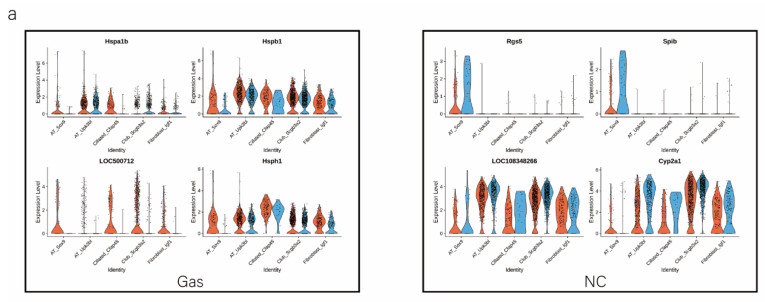
scRNA-seq reveals DEGs in AECs from normal control (NC) and phosgene-induced CALI model (Gas) groups. (**a**) Violin plots of the upregulated and downregulated DEGs in each cell cluster from the NC (blue violin) and Gas (red violin) groups. (**b**) Heatmap of the marker genes contributing from each cell subset. (**c**) Volcano plot of upregulated (red circles) and downregulated (blue circles) DEGs in SOX9-positive AECs from the Gas group compared with those in the SOX9-positive AECs from the NC group. GO term enrichment analysis of the (**d**) cellular component (CC), (**e**) biological process (BP), and (**f**) molecular function (MF) of the DEGs in NC and Gas group cells. (**g**) KEGG pathway analysis shows that the DEGs are mostly enriched in the PI3K-Akt signaling pathway. Heatmap of row-scaled expression of (**h**) signaling pathways regulating the pluripotency of stem cell genes and (**i**) the cell cycle for each averaged cluster.

**Table 1 ijms-24-00277-t001:** Annotation of pulmonary epithelial cell subtypes based on the maker gene.

Cluster	Gene Symbol	*p*_val	Avg_logFC	*p*_val_adj	Cell Type
AT_Sox9	*Sox9*	2.05 × 10^−177^	1.774493058	4.77 × 10^−173^	Alveolar epithelial progenitor
AT_Sox9	*Sh2d6*	2.49 × 10^−242^	2.847097	5.79 × 10^−238^	Alveolar epithelial progenitor
AT_Sox9	*Dclk1*	4.18 × 10^−209^	1.60923	9.70 × 10^−205^	Alveolar epithelial progenitor
AT_Sox9	*Rprml*	1.12 × 10^−208^	1.156296	2.61 × 10^−204^	Alveolar epithelial progenitor
AT_Sox9	*Gnat3*	5.50 × 10^−185^	1.430835	1.28 × 10^−180^	Alveolar epithelial progenitor
AT_Upk3bl	*Upk3bl*	0	1.890995855	0	Alveolar epithelial
AT_Upk3bl	*Krt15*	2.08 × 10^−258^	1.591570036	4.83 × 10^−254^	Alveolar epithelial
AT_Upk3bl	*Anxa1*	2.12 × 10^−206^	1.258136024	4.92 × 10^−202^	Alveolar epithelial
AT_Upk3bl	*Ppl*	8.32 × 10^−201^	1.038700757	1.93 × 10^−196^	Alveolar epithelial
AT_Upk3bl	*Cyp3a9*	1.69 × 10^−199^	1.044700287	3.94 × 10^−195^	Alveolar epithelial
Cillated_Cfap45	*Dynlrb2*	0	2.83119973	0	Ciliated epithelial
Cillated_Cfap45	*Aurkb*	0	2.474439543	0	Ciliated epithelial
Cillated_Cfap45	*Cfap126*	0	2.413891363	0	Ciliated epithelial
Cillated_Cfap45	*Ccdc153*	0	2.311122919	0	Ciliated epithelial
Cillated_Cfap45	*RGD1565611*	0	2.139924169	0	Ciliated epithelial
Club_Scgb3a2	*Scgb1a1*	0	1.590536	0	Club
Club_Scgb3a2	*Bpifb1*	1.07 × 10^−229^	1.377248	2.48 × 10^−225^	Club
Club_Scgb3a2	*Clca1*	3.03 × 10^−219^	0.915872	7.04 × 10^−215^	Club
Club_Scgb3a2	*Gp2*	7.42 × 10^−204^	1.547265	1.72 × 10^−199^	Club
Club_Scgb3a2	*Sftpd*	1.92 × 10^−182^	0.78207	4.47 × 10^−178^	Club
Fibroblast_Igf1	*Ptprc*	0	2.139825	0.167	Fibroblasts
Fibroblast_Igf1	*Igf1*	0	1.803506	0.092	Fibroblasts
Fibroblast_Igf1	*Cd53*	0	1.746076	0.121	Fibroblasts
Fibroblast_Igf1	*Apbb1ip*	0	1.633087	0.085	Fibroblasts
Fibroblast_Igf1	*Igsf6*	0	1.62369	0.111	Fibroblasts

## Data Availability

Not applicable.

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
