# Peer review of "Single-Cell RNA-Sequencing Reveals Epithelial Cell Signature of Multiple Subtypes in Chemically Induced Acute Lung Injury"

_ijms, 2022, doi:10.3390/ijms24010277_

Round 1

Reviewer 1 Report

The paper proposes to investigate toxic gaz (phosgene) damage (CALI) and regeneration process on alveolar lung cells by using scRNAseq to decipher markers of AEC subpopulations (Cell type state...). Data that are experimental in nature, depict that a unique AeC cell type expand following rat gaz poisoning. These cells can be progenitors of alveolar cells during the repair process. Although the paper is difficult to follow because of the huge amount of information, it remain understandable and pleasant to read. Although more synthetic figures would have greatly help the reader. Results are scientifically sound and clearly discussed.

Author Response

We greatly appreciate the thoughtful suggestions and insights provided by you, which have enriched the manuscript and produced a more balanced and better account of the research. Thank you very much for your assistance; we look forward to hearing from you soon. 

Reviewer 2 Report

1. Images in Fig.1 are of very low resolution making it hard to interpret the results. Gene names cannot be seen in violin plots and heatmaps in other figures too.

2. It is not clear in scRNA-seq analyses that Sox9+ cells are from CALI rats.

3. Staining in Fig 3F and 3G appear to be non-specific.

Author Response

Dear reviewer:

Thank you for your valuable comments. We greatly appreciate the thoughtful suggestions and insights that you provided, which have enriched the manuscript and presented a more balanced account of the research. We have included detailed responses to your comments below. We look forward to hearing from you soon.

  1. Images in Fig.1 are of very low resolution making it hard to interpret the results. Gene names cannot be seen in violin plots and heatmaps in other figures too.

Response: Thank you for your valuable comment. We provided low-resolution figures in the first draft of the manuscript to adhere to the journal guidelines regarding figures in initial submissions. As per your suggestion, we have provided clearer images with better quality in the revised manuscript.

  1. It is not clear in scRNA-seq analyses that Sox9+ cells are from CALI rats.

Response: Thank you for your valuable comment. Here, we analyzed the various cell subpopulations in alveolar lavage fluid (BALF) in normal controls and a CALI model using single-cell sequencing, identifying the proliferation of Sox9-positive AEC2 cell type. In animal models, we further identified the proliferation of subpopulation of these cells doubly positive for Sox9 and Sftpc by immunofluorescence, which confirmed the results of the scRNA-seq analyses. In revised manuscript, we checked and modified the method section.

  1. Staining in Fig 3F and 3G appear to be non-specific.

Response: Thank you for your valuable comment. We provided low-resolution figures in the first draft of the manuscript regarding figures in initial submissions. As per your suggestion, we have provided clearer images with better quality and provided the positive and negative controls of the figures in the revised manuscript. We have also updated Figures 3F and 3G.

Thank you again for your valuable feedback. We look forward to hearing from you soon.

Round 2

Reviewer 2 Report

2. Authors show in in their immunofluorescence data that Sox9+ AEC2 are mainly in CALI animals but not found in normal. However, in single cell analyses, it is not clear if Sox9+ cells are only found after injury or they are also in normal rats.

3. Staining in 3F and 3G appear to be non-specific (not the resolution).

Author Response

Dear reviewer:

Thank you for your valuable comments. We greatly appreciate the thoughtful suggestions and insights that you provided, which have enriched the manuscript and presented a more balanced account of the research. We have included detailed responses to your comments below. We look forward to hearing from you soon.

2. Authors show in in their immunofluorescence data that Sox9+ AEC2 are mainly in CALI animals but not found in normal. However, in single cell analyses, it is not clear if Sox9+ cells are only found after injury or they are also in normal rats.

Response: Thank you for your thoughtful critique of the work, we are very sorry for haven’t explained clearly. Actually, Sox9-positive cells are present both under the homeostasis and CALI state in the lung tissue, and could proliferate significantly in the state of lung injury. As shown in Fig. 1g, 2a and 2d, the Sox9+ cells exist both in normal state and CALI state in the sc-RNA data. In the models of immunofluorescence data, Fig. 3b confirmed the sequencing result, which positive for both SOX9 and SFTPC were highlighted with white arrows (line 215-216). We are very sorry for the unclear explanation, and we have marked it in the 2nd revised manuscript.

3. Staining in 3F and 3G appear to be non-specific (not the resolution).

Response: We are very sorry for not giving a satisfactory explanation. In the 2nd revised manuscript, we selected the positive and negative controls of their antibodies. As you can see in the figure below, we replaced Figure 3F (Gas, right) and 3G (NC, left). We believe these figures could be more clearly to distinguish the difference between NC and Gas group. Thank you very much for your advice.

Thank you again for your valuable feedback. We look forward to hearing from you soon.

Round 3

Reviewer 2 Report

The authors have not addressed either of the concerns even after 2 rounds of revision.

Author Response

Dear reviewer:

Thank you for your valuable comments and we deeply apologized for not addressing the comments clearly. We greatly appreciate the thoughtful suggestions and insights that you provided, which have enriched the manuscript and presented a more balanced account of the research. We have carefully modified the methods and result in the revision, checked the manuscript to make sure all references relevant to the contents of the manuscript, and included detailed responses to your comments below. We look forward to hearing from you soon.

1. Authors show in in their immunofluorescence data that Sox9+ AEC2 are mainly in CALI animals but not found in normal. However, in single cell analyses, it is not clear if Sox9+ cells are only found after injury or they are also in normal rats.

Response: Thank you for your thoughtful critique of the work, we deeply apologized that we haven’t explained clearly.

Sox9-positive cells are present both under the homeostasis and CALI state in the lung tissue, which could proliferate significantly during the state of chemically induced lung injury (CALI). In single-Cell RNA-Sequencing process, it can be found that the Sox9+ cells indeed exist in normal and CALI state. As shown in Fig. 2d, the quantity is about 50-80 in normal but 100-200 in CALI state, which indicated that the Sox9+ cells have proliferated.

In the models of immunofluorescence data, Fig. 3b confirmed the sequencing result, which positive for both SOX9 and SFTPC were highlighted with white arrows. We are very sorry for the unclear explanation.

2. Staining in 3F and 3G appear to be non-specific (not the resolution).

Response: We are very sorry for not giving a satisfactory explanation. In revised manuscript, we carefully analyzed and selected more appropriate pictures for replacement (Figure 3f and 3g). We believe these figures could be more clearly to distinguish the difference between NC and Gas group. Thank you very much for your advice.

Thank you again for your valuable feedback. We look forward to hearing from you soon.